# Association of nasopharyngeal *Dolosigranulum pigrum* and *Corynebacterium* species with post-acute sequelae of SARS-CoV-2 in a longitudinal cohort

Bradley Ward,[1,2] Laure B. Bindels,[3,4] Jean-Luc Balligand,[4,5,6] Bertrand Bearzatto,[7] Guido Bommer,[8] Patrice D. Cani,[3,4,9,10] Julien De Greef,[2,11] Joseph P. Dewulf,[2,12,13] Laurent Gatto,[14] Vincent Haufroid,[2,12] Sébastien Jodogne,[15] Benoît Kabamba,[9,16] Sébastien Pyr dit Ruys,[2] Didier Vertommen,[8] Jean Cyr Yombi,[9,11] Leïla Belkhir,[2,11] Laure Elens[1,2]

**ABSTRACT** This longitudinal study investigated the differential composition of the nasopharyngeal microbiome in patients presenting different COVID-19 infectious phenotypes and its evolution during convalescence, with a focus on post-acute sequelae of SARS-CoV-2 (PASC) and its potential microbiome-related mechanisms. Microbiota composition was assessed for a cohort of healthy participants ($n = 25$), influenza patients ($n = 24$), and patients with moderate ($n = 50$) and severe ($n = 57$) COVID-19. Samples were collected at two time points: during the acute infection phase and at approximately 3-month follow-up. From collected nasopharyngeal swab samples, metagenomics using shotgun sequencing was performed and the microbiota composition was analyzed. Alpha and beta diversity analyses revealed no significant differences in overall community diversity between patient groups across visits. However, differential abundance testing identified specific species, such as *Dolosigranulum pigrum* and various *Corynebacterium* species, whose profiles correlated with PASC development. Furthermore, the analysis of microbial co-associations identifies commensal species, including *D. pigrum* and *Corynebacterium* species, which are less abundant in patients who develop PASC, consistent with a potential protective role suggested by experimental studies but not proven by our observational data. Antibiotic use was associated with lower levels of key protective taxa, which may increase susceptibility to PASC in case of superinfection. These findings highlight the potential importance of the nasopharyngeal microbiome in acute COVID-19 disease outcomes and suggest that preserving or restoring a balanced respiratory microbiome could mitigate the risk of COVID-19 persistent symptoms and PASC development. Our results may set the stage for future clinical interventions involving probiotics or microbial-derived metabolites to promote respiratory health post-COVID-19.

**IMPORTANCE** This study highlights the importance of bacteria naturally found in the upper respiratory tract, particularly the nasopharynx (the nasopharyngeal microbiome), in shaping how severely COVID-19 affects patients and whether they experience persistent symptoms, also called long-COVID or post-acute sequelae of SARS-CoV-2 (PASC). By examining microbiome samples from healthy people, influenza patients, and individuals with COVID-19 during acute and convalescent phases, we found that certain commensal bacteria, namely, *Dolosigranulum pigrum* and *Corynebacterium* species, were less abundant in individuals who developed long-COVID and more abundant in those who fully recovered. We also observed that antibiotic treatment was associated with lower abundances of these commensal taxa, in turn coinciding with a higher frequency of PASC. These findings suggest that the composition of the nasopharyngeal microbiome is associated with recovery trajectories after COVID-19 and motivate future research

**Peer Reviewer** Cindy M. Liu, George Washington University Milken Institute School of Public Health, Washington, DC, USA

Address correspondence to Laure Elens, laure.elens@uclouvain.be.

Leïla Belkhir and Laure Elens contributed equally to this article. Author order was determined on the basis of seniority to this project.

P.D.C. is the inventor on patent applications dealing with the use of specific bacteria and components in the treatment of different diseases. P.D.C. was co-founder of Enterosys.

See the funding table on p. 12.

into treatments aimed toward the microbiome to improve respiratory health following infection.

**CLINICAL TRIALS** This study is registered with ClinicalTrials.gov as NCT05557539.

**KEYWORDS** post-acute sequelae of COVID19, COVID-19, respiratory microbiome, *Dolisigranum pigrum*, *Corynebacterium*

The human upper respiratory tract, particularly the nasopharynx, hosts a diverse microbiome essential to health and disease prevention. In healthy adults, the nasopharyngeal (NP) microbiome is typically dominated by genera such as *Corynebacterium*, *Dolosigranulum*, *Moraxella*, *Streptococcus*, and *Haemophilus*; a balanced community is thought to contribute to mucosal immune maturation and colonization resistance against respiratory pathogens (1, 2). Disruption of this microbiome, associated with respiratory infections, asthma, or chronic obstructive pulmonary disease, can impair respiratory clearance and exacerbate disease progression (3). Severe acute respiratory syndrome coronavirus 2 (SARS-CoV-2), the virus responsible for COVID-19, primarily infects the nasal and nasopharyngeal epithelium (4). COVID-19 symptoms range from asymptomatic cases to severe respiratory failure and death, in addition to persistent symptoms, known as post-acute sequelae of SARS-CoV-2 (PASC) infection. PASC can persist for weeks or months after COVID-19 and develops in between 34% and 54% of cases (5). Symptoms frequently include fatigue, cognitive impairment ("brain fog"), dyspnea, and chest pain (5, 6).

Existing studies investigating the impact of SARS-CoV-2 on the nasopharyngeal microbiome have produced inconsistent results; some found significant microbiome disruptions associated with increased disease severity, while others reported minimal differences compared to healthy controls (7–12). Most studies utilized 16S rRNA sequencing, with only two employing comprehensive shotgun sequencing, both involving small cohorts (37–40 patients). Furthermore, most studies only focused on the acute disease, without examining longitudinal microbiome evolution, and so there is a notable gap in understanding how the nasopharyngeal microbiome evolves after acute infection or its relationship with persistent symptoms, leaving a critical knowledge gap regarding the microbiome's role in PASC. This study addresses these gaps by examining longitudinal changes in the nasopharyngeal microbiome of COVID-19 patients during acute infection and at 3-month follow-up using shotgun sequencing. We compare microbiome profiles of COVID-19 patients with different clinical severities to healthy individuals and patients hospitalized for influenza-induced acute respiratory failure. By correlating microbiome dynamics with clinical outcomes, we aim to clarify potential associations between microbial shifts, disease severity, and persistent symptoms. Ultimately, this research aims to deepen understanding of the nasopharyngeal microbiome's involvement in COVID-19 and PASC, informing strategies for prevention and management of ongoing symptoms.

## RESULTS

### Study cohort

The cohort's mean age was 53.3 years, with 53.2% male and 46.8% female. Average body mass index (BMI) was 26.4. Ethnicities included 77.6% Caucasian, 10.3% African, and the rest Arabic, Asian, Latino-American, Haitian, or unknown. Around 80% were fully vaccinated against COVID-19 before the study. Severe COVID-19 and PASC patients had a higher mean age, a higher proportion of male patients, a higher mean BMI, a higher prevalence of smoking and drinking, and a lower prevalence of full vaccination than those with mild cases or full recovery. The main core taxonomic profiles of patients (species found in all patients) included *Klebsiella pneumoniae*, *Corynebacterium amycolatum*, *Escherichia coli*, *Paracidovorax avenae*, *Bacillus paralicheniformis*, *Bacillus*

*amyloliquefaciens*, and *Salmonella enterica* in order of mean relative abundance. More details can be found in Table S1, alongside information regarding sample taxonomy profiles (prevalence and relative abundances) in Fig. S1.

## Diversity

Nasopharyngeal alpha diversity was evaluated with observed richness and Shannon diversity. At both visit 1 and visit 2, the Kruskal-Wallis test revealed no significant differences in diversity in either the observed richness (visit 1: $H = 6.46$, df = 3, $P = 0.091$; visit 2: $H = 2.24$, df = 3, $P = 0.524$) or the Shannon diversity (visit 1: $H = 4.50$, df = 3, $P = 0.212$; visit 2: $H = 4.36$, df = 3, $P = 0.225$) between any of the patient groups (healthy, flu, mild COVID, and severe COVID). Subsequently, Wilcoxon rank-sum tests revealed no significant differences between recovered and PASC patients at either visit in observed richness (visit 1: $W = 1244.5$, $P = 0.45$; visit 2: $W = 730.5$, $P = 0.51$) or Shannon diversity (visit 1: $W = 1287$, $P = 0.29$; visit 2: $W = 793$, $P = 0.19$). And finally, there were no significant pairwise differences observed when comparing data from visit 1 to visit 2 for any comparator groups (healthy, flu, mild COVID, severe COVID, recovered patients, and PASC patients).

Although there is overlap, principal coordinates analysis (PCoA) with permutational multivariate analysis of variance (PERMANOVA) shows significant associations between microbiota composition and cohort groupings at visit 1 (healthy, flu, mild COVID, and severe COVID) (Fig. 1). When cohort groupings were modeled alone, the model explained 3.8% of the variance (model: *P*-value = 0.007). When cohort groupings and sex were modeled together, the explained model variance increased to 5.0% (model: *P*-value = 0.005; grouping: variance explained = 3.5%, *P*-value = 0.045; sex: variance explained = 1.2%, *P*-value = 0.083).

Permutational analysis of multivariate dispersions (PERMDISP) showed no significant differences, for cohort groupings or sex, indicating our results are unlikely driven by unequal group dispersions. Other participant variables (ethnicity, age, BMI, smoking/drinking status, SARS-CoV-2 vaccinations, and PASC development) were also

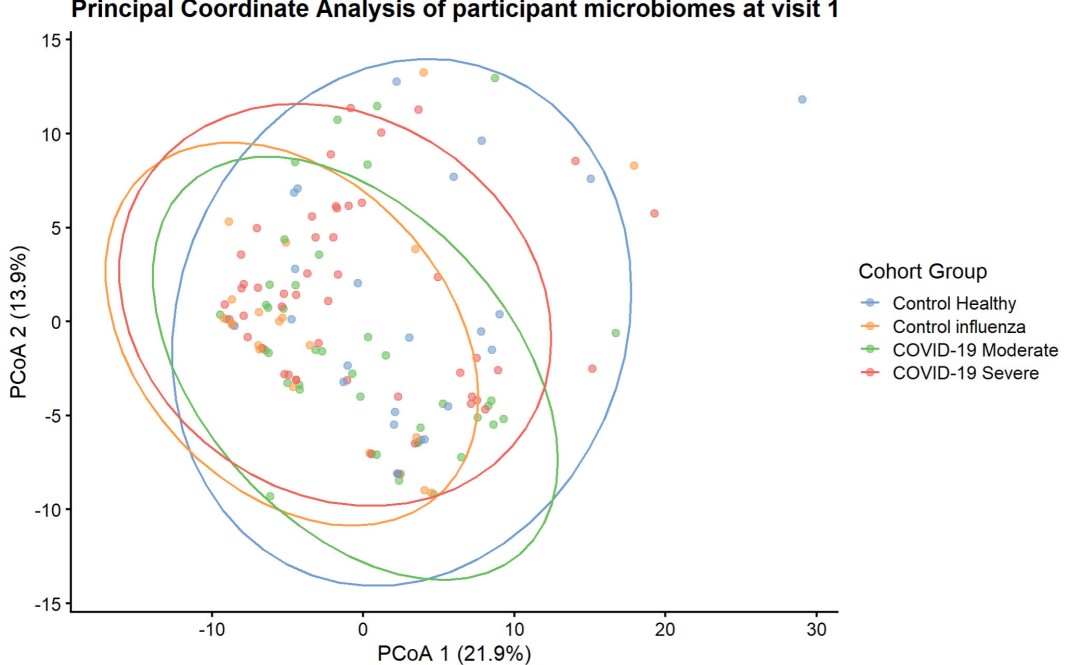

**FIG 1** Beta diversity analysis between cohort disease groups at visit 1. PCoA of participant microbiomes during visit 1 (acute phase) based on Aitchison distance, corresponding to Euclidean distances between centered log-ratio (CLR) transformed abundance values. Samples are colored according to cohort grouping and overlaid with 95% confidence ellipses.

assessed but did not correlate significantly with the microbiota composition. The same grouping variables were also assessed at visit two but did not reveal any significant association with microbiota composition.

## Differential abundance analysis

Differential abundance testing was conducted using analysis of compositions of microbiomes with bias correction 2 (ANCOM-BC2); sex, antibiotic treatment, and age were also included in the model to account for confounding. Species abundance was modeled against COVID-19 disease severity (moderate vs severe) at visit 1. Two species, *Corynebacterium tuberculostearicum* (LogFC = −1.82, *q*-val = 0.026) and *Malassezia restricta* (LogFC = 1.40, *q*-val = 0.0051) were differentially abundant in severe COVID-19 patients in comparison to mild COVID-19 patients. *M. restricta* was also significantly more prevalent in severe COVID-19 patients, further associating with increased disease severity (Fisher's exact test, *P*-value = 0.0021, odds ratio [OR] = 6.5).

The second analysis tested for differentially abundant species at visit 1 between COVID-19 patients who developed PASC at follow-up (visit 2) and those that fully recovered (Fig. 2); COVID-19 severity, sex, antibiotic treatment, and age were included within the model. The analysis highlighted four significant species: *Corynebacterium accolens* ($q = 4.18^{e−06}$, LogFC = 1.35), *Dolosigranulum pigrum* ($q = 1.42^{e−06}$, LogFC = −3.98), *Staphylococcus epidermidis* ($q = 4.39^{e−05}$, LogFC = 1.40), and *Corynebacterium propinquum* ($q = 6.08^{e−03}$, LogFC = −1.54). Of these, only *C. accolens* passed sensitivity analysis. However, *C. accolens* was significantly less prevalent in patients that developed PASC (Fisher's exact test, *P*-value = 0.0047, odds ratio = 0.19). *D. pigrum* and *C. propinquum* were also slightly less prevalent in patients that developed PASC, but not significantly (*D. pigrum*: Fisher's exact test, *P*-value = 0.11, odds ratio = 0.25; *C. propinquum*: Fisher's exact test, *P*-value = 0.50, odds ratio = 0.52).

Differentially abundant species between PASC patients and recovered patients were also assessed at visit 2 (Fig. 3). At visit 2, four species were found to differ significantly in their abundance: *C. propinquum* ($q = 2.07^{e−04}$, LogFC = 2.43) and *Cutibacterium*

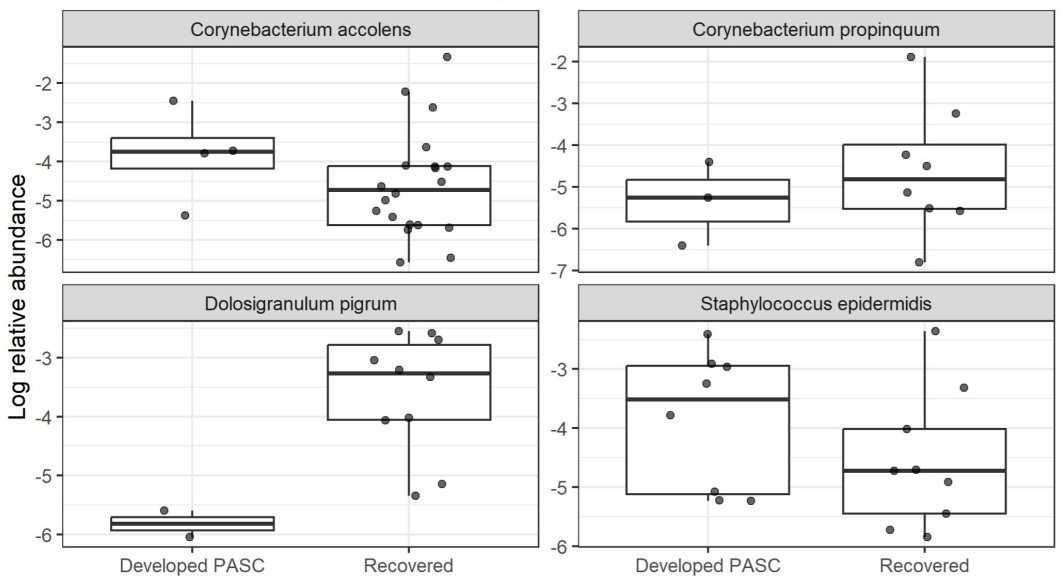

**FIG 2** Differentially abundant species at visit 1 between PASC patients and recovered patients. Boxplots with jittered data points of the log-transformed relative abundance of microbial species that are significantly differentially abundant (*q*-value ≤ 0.05 and LogFC ≥ 1) in COVID-19 patients at visit 1 between patients that recovered and patients that developed PASC at visit 2, as determined via ANCOM-BC2 analysis after controlling for disease severity, sex, and antibiotic treatment before recruitment.

*granulosum* ($q = 2.07^{e-04}$, LogFC = 1.22) were more abundant in PASC, while *Corynebacterium pseudodiphtheriticum* ($q = 1.42^{e-05}$, LogFC = −1.73) and *C. accolens* ($q = 5.54^{e-03}$, LogFC = −1.68) were less abundant. *D. pigrum* was also at significantly lower abundance in PASC patients, but not by a large magnitude ($q = 5.54^{e-03}$, LogFC = −0.69). None passed sensitivity analysis.

A paired analysis was performed to assess longitudinal evolution of patient microbiomes in those patients that recovered vs those patients that developed PASC (controlling for antibiotic exposure, age, sex, and COVID-19 severity), but no significantly differentially abundant species were discovered. A similar analysis was also done for COVID severity (mild COVID vs severe COVID, controlling for antibiotic exposure, age, and sex), but nothing was identified.

To assess antibiotic use on the respiratory microbiome, a comparison was performed between hospitalized patients (severe flu and severe COVID-19 patients) who received antibiotic therapies before biological samples were collected at visit 1 (roughly 18% of severe COVID-19 patients and 42% of severe flu patients) and those who did not (Fig. 4). On average, antibiotic therapy was given 3 days before sample collection. *D. pigrum* ($q = 5.60^{e-05}$, LogFC = −2.25), *Cutibacterium acnes* (*C. acnes*) ($q = 4.25^{e-04}$, LogFC = −1.01), *C. tuberculostearicum* ($q = 4.90^{e-03}$, LogFC = 1.07), *Streptococcus pneumoniae* ($q = 4.82^{e-02}$, LogFC = 1.45), and *C. accolens* ($q = 1.45^{e-02}$, LogFC = −1.09) were all found at significantly lower abundance in antibiotic-treated patients. None of these species passed sensitivity analysis. Fisher's exact test was also conducted to assess species presence/absence, but none were evaluated as significant.

To investigate long-term antibiotic use, we ran a longitudinal analysis from visit 1 to visit 2 using the same sub-cohort of hospitalized patients, between those that took antibiotics and those that did not, but no differences were found.

COVID-19 severity was significantly associated with PASC at visit 2, with severe compared to mild acute COVID-19 showing higher odds of PASC (OR 5.12, 95% confidence interval [CI] 1.77–15.83, $P = 0.001$). Given that antibiotic-associated alterations in the respiratory microbiome, including *D. pigrum* and *C. accolens*, were linked to PASC in the differential abundance analysis, we hypothesized that antibiotic exposure

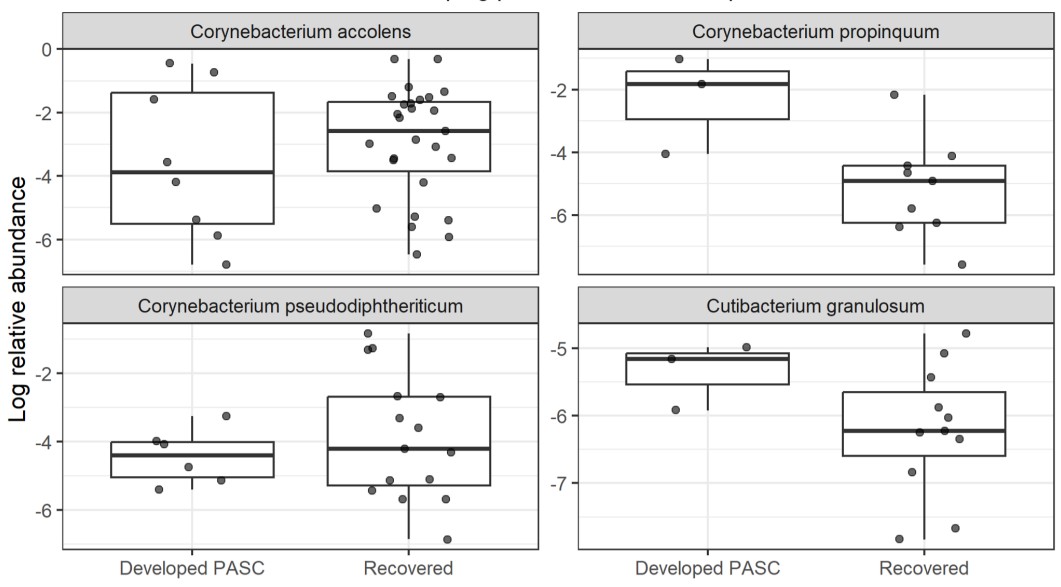

**FIG 3** Differentially abundant species at visit 2 between PASC patients and recovered patients. Boxplots with jittered data points of the log-transformed relative abundance of species that are significantly differentially abundant ($q$-value ≤ 0.05 and LogFC ≥ 1) at visit 2 between COVID patients who developed PASC versus those who recovered, as determined by an ANCOM-BC analysis controlling for antibiotic treatment and sex.

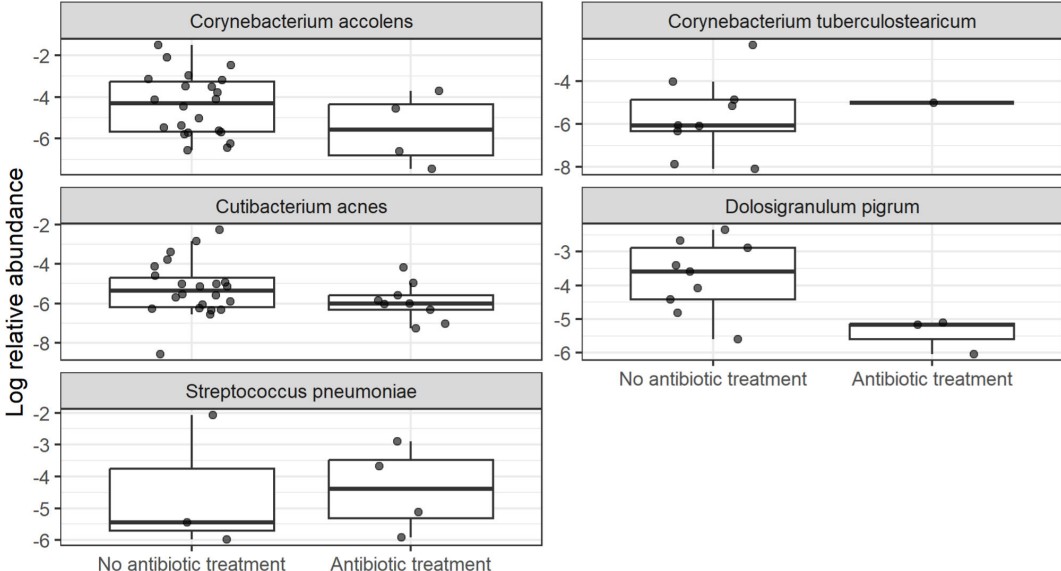

**FIG 4** Differentially abundant species at visit 1 between patients treated with or without antibiotics. Boxplots with jittered data points of the log-transformed relative abundance of species that are significantly differentially abundant ($q$-value $\leq 0.05$ and LogFC $\geq 1$) at visit 1 between hospitalized patients (severe COVID-19 and influenza control patients) who received antibiotic treatment before sample collection and those that did not. Analysis was run using ANCOM-BC2 while controlling for sex and infection type (COVID vs influenza).

during acute infection might also be associated with subsequent PASC. Because only hospitalized (severe) patients received antibiotics, this analysis was restricted to severe COVID-19 cases. In this group, a greater proportion of patients receiving antibiotics during hospitalization developed PASC (9/10) than those who did not receive antibiotics (12/23). Fisher's exact test suggested higher odds of PASC among antibiotic-treated patients (OR 7.80, 95% CI 0.84–392.92, $P = 0.054$), although this association did not reach conventional statistical significance and the confidence interval was wide, reflecting the small sample size.

### Microbiome association network

We next examined significant co-abundance relationships between prevalent nasopharyngeal species using a SparCC-based association network (Fig. 5). Investigating this association network in the context of PASC, *D. pigrum*, which was identified through ANCOM-BC2 as being of potential relevance to the development of PASC during the acute phase, is seen to significantly positively associate with three *Corynebacterium* species, *C. accolens*, *C. pseudodiphtheriticum*, and *C. propinquum*. Continuing to explore the network, we can see that *C. accolens* appears to behave as a hub node, through which a number of negative correlations are seen with species such as *Escherichia coli*, *Staphylococcus aureus* (indirectly), *Corynebacterium amycolatum*, *Klebsiella pneumoniae*, *Acinetobacter jandaei*, *Neisseria gonorrhoeae*, and *Salmonella enterica*.

### DISCUSSION

Our study demonstrates that although overall diversity metrics remained largely unchanged between patients during both the acute phase and follow-up, differential abundance analyses reveal subtle yet significant shifts in key bacterial taxa that may influence PASC development. In particular, the contrasting abundance patterns of *D. pigrum* and various *Corynebacterium* species during acute illness suggest that

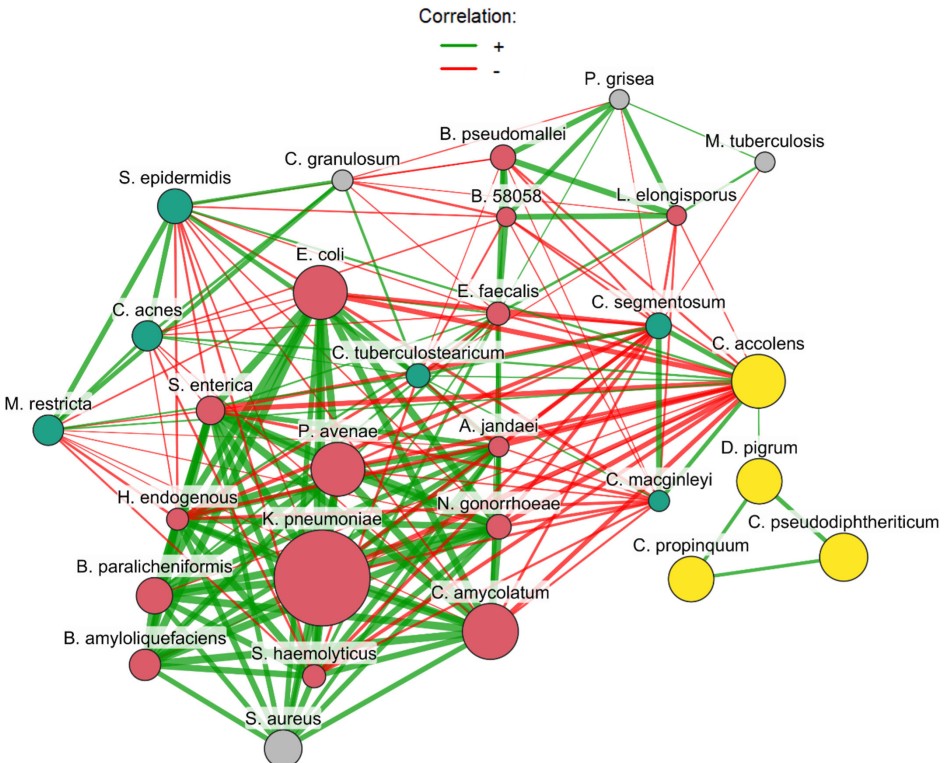

**FIG 5** SparCC co-abundance network of prevalent nasopharyngeal species. Each node represents a bacterial species, and edges represent significant co-abundance relationships across the cohort. Green edges indicate positive associations (species that tend to increase or decrease together), and red edges indicate negative associations (species that vary inversely). The layout (force-directed) reveals a commensal cluster comprising *Dolosigranulum pigrum* and several *Corynebacterium* species that is predominantly negatively connected to opportunistic or pathogenic taxa. Members of the commensal consortium associated with lower PASC frequency are colored yellow, taxa negatively associated with *C. accolens* are colored red, taxa positively associated with *C. accolens* are colored green, and taxa not directly correlated with *C. accolens* are colored gray. Node size is reflective of mean relative abundance of species when identified (larger node = more abundant) and edge thickness reflects absolute correlation strength.

nasopharyngeal microbiome composition may play a role not only in disease trajectory but also in long-term respiratory outcomes.

Differential abundance analyses revealed taxa significantly differing between patients who developed PASC and those who recovered. Specifically, *D. pigrum* and *Corynebacterium* species emerged as particularly salient. *D. pigrum* abundance was markedly reduced (−3.98 LogFC) during the acute phase in patients who later developed PASC. Conversely, *Corynebacterium* species exhibited more complex patterns. *C. accolens* was enriched during acute illness in patients progressing to PASC (1.35 LogFC) yet, paradoxically, was more prevalent overall among recovered individuals. It could be inferred that while *C. accolens* might be a marker of dysbiosis in some PASC cases (high abundance), its absence is the more common feature of PASC (low prevalence). Meanwhile, in these same patients, *C. propinquum* abundance was reduced (−1.54 LogFC), hinting at species-specific roles.

These findings align with a growing body of literature indicating genera such as *Prevotella*, *Staphylococcus*, and *Streptococcus* are associated with severe COVID-19, while commensals like *Dolosigranulum* and *Corynebacterium* associate with milder phenotypes (13). The commensals frequently dominate the nasopharyngeal microbiome, "gatekeeping" against respiratory pathogens by occupying niches and interacting with host immune cells (14–16). Alterations in this community can therefore compromise mucosal defenses and increase susceptibility to infections.

## *D. pigrum*

*D. pigrum*, a gram‑positive, facultatively anaerobic coccus, primarily inhabits the nasopharynx as a commensal (17). Its genomic stability and robust genetic defense systems underpin its status as a true commensal.

Murine studies reinforce this hypothesis. Pneumococcus-infected mice treated with *D. pigrum* exhibited early robust inflammatory response, with increased macrophage/neutrophil recruitment and elevated IL-1β, IFN-γ, and IL-6 levels, followed by a resolving phase marked by decreased neutrophils, TNF-α, CCL2, and IL-6, and increased regulatory cytokines (18). These effects required cooperation with alveolar macrophages, dendritic cells, and T cells. Similarly, *in vitro*, *D. pigrum* modulated IFN-β, CXCL8, CCL5, and CXCL10 production in SARS-CoV-2 challenged human respiratory epithelial cells, reducing viral replication and cell damage (19). One proposed mechanism of PASC involves residual lung damage following excessive inflammation in response to SARS-CoV-2 (20). Together with experimental evidence for immunomodulatory effects of *D. pigrum* and our observation of lower *D. pigrum* abundances in patients with PASC, this raises the hypothesis that higher *D. pigrum* levels during acute COVID-19 may modulate host immune responses to minimize lung damage, subsequently reducing PASC risk (although our data remain purely associative). Proposed mechanisms through which this modulation arises include maintaining mucosal homeostasis, inhibiting pathogen growth, reducing local inflammation, and/or promoting balanced cytokine responses, thus suggesting its potential as a probiotic (21, 22).

Altogether, murine studies and our observations suggest that maintaining high levels of *D. pigrum* during acute COVID-19 may limit immune-mediated lung injury, reducing PASC risk.

## *Corynebacterium* spp.

*Corynebacterium* species appear to play complex roles in respiratory health, with our data indicating species-specific and sometimes opposing effects. Regarding acute disease severity at visit 1, *C. tuberculostearicum* levels were significantly lower in severe COVID-19 patients compared to moderate cases. In the context of PASC, *C. propinquum* displayed a dynamic shift: it was initially reduced in patients who developed PASC but increased in abundance during follow-up (visit 2). By contrast, both *C. pseudodiphtheriticum* and *C. accolens* were found at higher abundances in recovered individuals during follow-up compared to PASC patients. *C. accolens* specifically exhibited a complex profile; while it was more prevalent overall among recovered individuals, its relative abundance was enriched during the acute phase in patients who later developed PASC.

Immunological insights help explain these findings. Some species (*C. pseudodiphtheriticum*) are efficiently internalized by alveolar macrophages, triggering cytokine production and enhancing antigen-specific responses (23). Additionally, *C. accolens* produces lipases leading to free fatty acid release, which possess potent anti-pneumococcal activity, potentially reducing secondary infections (24, 25). Particularly relevant to COVID-19, certain *Corynebacterium* strains, notably those belonging to *C. accolens*, also downregulate SARS-CoV-2 entry proteins, potentially limiting infection and replication (26).

## The protective consortium

The nasopharyngeal microbial co-abundance network (Fig. 5) reveals a complex web of interactions that may underlie the observed associations between this microbiome and PASC. Notably, *D. pigrum* positively correlated with multiple *Corynebacterium* species (*C. accolens*, *C. pseudodiphtheriticum*, *C. propinquum*), all associated with reduced PASC development. This correlation is supported by the observation that *D. pigrum* is likely dependent on *Corynebacterium* for nutrients and colonization support (27). Such consortia may stabilize mucosal integrity, block pathogen overgrowth that worsens

COVID-19-related lung damage, and/or reduce SARS-CoV-2 susceptibility through ACE2 downregulation and binding inhibition (24, 26, 28, 29).

Equally notable is the strong negative correlation between *C. accolens* and pathogens like *K. pneumoniae*, a respiratory pathogen responsible for one in six ICU coinfections in COVID-19 patients (30, 31). Likewise, *Acinetobacter* species are also known opportunistic respiratory pathogens in hospital settings (32, 33). The strong negative correlations could imply that commensals like *C. accolens* may inhibit pathogen overgrowth through the mechanisms discussed above. Such pathogen inhibition is observed *in vivo* and *in vitro*, whereby *D. pigrum* and *Corynebacterium* prevent *Staphylococcus aureus* and *Streptococcus pneumoniae* colonization (27, 34).

The network connectivity further supports these taxa as potential biomarkers of disease severity and recovery trajectories. A commensal axis associated with favorable outcomes involving *D. pigrum* and *Corynebacterium* species could indicate microbiome resilience, reducing SARS-CoV-2 inflammation and preventing superinfection. Conversely, disruption of this consortium could predispose patients to inflammatory-related lung injury and subsequent PASC development.

## Antibiotic use as a predictor of PASC

An important study finding was the impact of antibiotic use on the respiratory microbiome. Hospitalized COVID-19 patients receiving antibiotics showed higher likelihood of developing PASC compared to untreated patients (OR 7.80, $P = 0.054$). Although marginally outside conventional statistical significance, it is plausible that patients prescribed antibiotics for suspected bacterial coinfections suffered enhanced lung damage due to the combined effects of viral and bacterial infections, increasing PASC risk. Pulmonary coinfections in COVID-19 patients lead to poorer clinical outcomes and more severe lung abnormalities (35–37). These severe lung abnormalities suffered during acute COVID-19 and their ongoing persistence are major predictors of PASC development (38–40).

Antibiotics, while essential therapeutically, can disturb the microbiome. In our study, antibiotic-prescribed patients exhibited reduced abundances of candidate commensal taxa associated with favorable outcomes (*D. pigrum* and *C. accolens*), suggesting two hypotheses. First, antibiotics directly reduce beneficial microbes, impairing protection and increasing PASC risk. Second, patients with initially low levels of commensals might be predisposed to infections, prompting antibiotic use. The first scenario underscores the importance of prioritizing narrow-spectrum agents, whenever clinically feasible, to maintain microbiome integrity. Additionally, co-therapy with respiratory probiotics during/after antibiotic therapy may help minimize beneficial commensal reduction. The latter scenario highlights microbiome composition as a predictor of patient outcomes and the protective role of respiratory probiotics.

## Limitations

Despite the depth of our analyses, several limitations remain. First, the overall sample size, data sparsity, and the inherent variability in nasopharyngeal sampling may have constrained our ability to detect clearer microbial shifts, and although confounders such as sex, age, disease severity, and antibiotic exposure were controlled for, residual confounding is possible. Second, while we identified several key taxa associated with PASC, not all findings passed sensitivity analyses. Moreover, the potential of using these taxa associated with favorable outcomes as biomarkers for disease prognosis warrants further investigation. Early identification of patients at risk for PASC based on their nasopharyngeal microbial profiles could facilitate timely, personalized interventions aimed at restoring microbial balance and mitigating lung damage. Furthermore, although blank controls were not included, we minimized contamination and bias by processing Zymo balanced cell controls alongside study samples (with aliquots from the same batch used throughout extraction and sequencing). Potential contaminants were minimized by employing stringent Kraken‐based species identification combined

with a Bracken minimum‐read cutoff of 200 per feature to eliminate low‐abundance misidentifications. Additionally, contaminant species were identified using the frequency method from decontam, which leverages batch information and DNA concentration.

## MATERIALS AND METHODS

### Study cohort

The study used samples from participants enrolled in the HYGIEIA study between 2020 and 2024: 25 healthy controls, 24 influenza patients, 50 moderate COVID-19 patients, and 57 severe COVID-19 patients (41). All participants provided their informed consent before participation in the study. Nasopharyngeal (NP) swab samples were obtained at two time points: first, at diagnosis/study inclusion, and then during a follow-up visit, on average 81 days later. Around 80% of participants continued until the follow-up. COVID severity was graded according to World Health Organization Clinical Progression Scale scores, while PASC status at the follow-up visit was defined as the presence or continuation of symptoms at visit 2 without any obvious alternative explanations.

### DNA extraction, library preparation, and sequencing

DNA extraction was conducted using the ZymoBIOMICS DNA Miniprep Kit (Catalog No. D4300, Zymo Research, USA), according to the manufacturer instructions with minor modifications. Briefly, the NP swab tip was cut into the kit's lysis tube, and 900 µL sample media plus 100 µL DNA/RNA Shield was added. Tubes were vortexed at maximum speed for 45 minutes, centrifuged at $12,000 \times g$ for 1 minute, and supernatant was transferred to the purification columns, and the ZymoBIOMICS DNA Miniprep Kit protocol was then followed from this point forward.

Library preparation was conducted on 25–49 ng of DNA using an Illumina DNA Prep kit (Catalog No. 20018704, Illumina, USA) and following manufacturer's instructions. Pooled libraries (17-plex) were sequenced on the Illumina NextSeq 1000 platform using a NextSeq 1000/2000 P2 Flow Cell.

### Bioinformatics and statistical analysis

Reads were trimmed with Trimmomatic (v0.39) and filtered using Bowtie2 (v2.4.5) to remove human-aligned sequences (GRCh38); low-read samples (<500,000 remaining) were removed from downstream analyses. Taxonomic classification was performed using Kraken2 (v2.1.3) with a RefSeq database, followed by abundance estimation using Bracken (v2.7).

Alpha diversity was assessed using two complementary indices: observed richness and Shannon diversity. Comparisons across patient groups were performed using Wilcoxon or Kruskal-Wallis tests when appropriate. Beta diversity was analyzed using Aitchison distance, and PERMANOVA quantified its association to external factors. Differential abundance was analyzed using ANCOM-BC2 with relevant confounding factors (e.g., cohort grouping, sex, and antibiotic treatment) incorporated into the model.

To infer global co-abundance patterns between species, we first restricted the data set to species present in at least 10% of all samples (all patient groups, visit 1 and visit 2). We then computed SparCC correlations using species raw counts (SpiecEasi::sparcc). Statistical support for each correlation was assessed using 1,000 bootstrap resamples, and *P*-values were adjusted for multiple testing using the Benjamini-Hochberg procedure. We retained taxon correlation pairs with false discovery rate < 0.01 and an absolute correlation coefficient $|r| \geq 0.3$ to construct an undirected co-abundance network. Nodes represent species and edges represent significant correlations; positive correlations are depicted in green and negative correlations in red.

More detail for methodology can be found in the supplemental material.

## ACKNOWLEDGMENTS

We sincerely thank all patients who participated in this clinical study. We are also grateful to the clinical research coordinators at Cliniques universitaires Saint Luc for their indispensable assistance in patient recruitment and sample collection.

This research was financially supported by the Sofina COVID Solidarity Fund, administered by the King Baudouin Foundation, initiated by the Fondation Saint-Luc (grant number 2021-I4201010–221801), and the FNRS Urgent Research Credit (CUR: HC01020F). L.B.B. is a Collen-Francqui Research Professor and grateful for the support of the Francqui Fondation. P.D.C. is a recipient of grants from ARC-FSR (ARC19/24–096) and FNRS (FNRS T.0032.25, WELBIO-CR-2022A-02P, and EOS 40007505). B.W. is the recipient of an FRC starting grant (Promotor Leïla Belkhir) and an FSR grant (Promotor Laure Elens). The funding agencies had no role in study design, data collection and interpretation, or the decision to submit the work for publication.

## AUTHOR AFFILIATIONS

[1]UCLouvain [Bruxelles-Woluwe], Louvain Drug Research Institute (LDRI), Integrated Pharmacometrics, Pharmacogenomics and Pharmacokinetics Group (PMGK), Woluwe-Saint-Lambert, Belgium

[2]UCLouvain [Bruxelles-Woluwe], Institut de Recherche Expérimentale et Clinique (IREC), Louvain Center for Toxicology and Applied Pharmacology (LTAP), Woluwe-Saint-Lambert, Belgium

[3]UCLouvain [Bruxelles-Woluwe], Louvain Drug Research Institute (LDRI), Metabolism and Nutrition Research Group (MNUT), Woluwe-Saint-Lambert, Belgium

[4]WELBIO (Walloon Excellence in Life Sciences and Biotechnology), WELBIO Department, WEL Research Institute, Wavre, Belgium

[5]UCLouvain [Bruxelles-Woluwe], Institut de Recherche Experimentale et Clinique (IREC), Pole of Pharmacology and Therapeutics (FATH), Woluwe-Saint-Lambert, Belgium

[6]Cliniques universitaires Saint-Luc, Woluwe-Saint-Lambert, Belgium

[7]Institut de Recherche Experimentale et Clinique (IREC), Centre for Applied Molecular Technologies (CTMA), UCLouvain, Woluwe-Saint-Lambert, Belgium

[8]De Duve Institute, MASSPROT, UCLouvain, Woluwe-Saint-Lambert, Belgium

[9]UCLouvain [Bruxelles-Woluwe], Institut de Recherche Experimentale et Clinique (IREC), Woluwe-Saint-Lambert, Belgium

[10]Section of Biomolecular Medicine, Division of Systems Medicine, Department of Metabolism, Digestion and Reproduction, Imperial College London, London, United Kingdom

[11]Department of Internal Medicine and Infectious Diseases, Cliniques universitaires Saint-Luc, Woluwe-Saint-Lambert, Belgium

[12]Department of Laboratory Medicine, Cliniques universitaires Saint-Luc, Bruxelles-Woluwe, Belgium

[13]UCLouvain [Bruxelles-Woluwe], de Duve Institute (DDUV), Department of Biochemistry, Woluwe-Saint-Lambert, Belgium

[14]UCLouvain [Bruxelles-Woluwe], de Duve Institute (DDUV), Computational Biology and Bioinformatics Unit (CBIO), Woluwe-Saint-Lambert, Belgium

[15]UCLouvain [Louvain-La-Neuve], Institute of Information and Communication Technologies, Electronics and Applied Mathematics (ICTEAM), Computer Science and Engineering Department (INGI), Louvain-la-Neuve, Belgium

[16]UCLouvain [Bruxelles-Woluwe], Institut de Recherche Experimentale et Clinique (IREC), Pôle de Microbiologie (MBLG), Woluwe-Saint-Lambert, Belgium

## AUTHOR ORCIDs

Bradley Ward  http://orcid.org/0000-0003-0778-0153
Jean-Luc Balligand  http://orcid.org/0000-0002-0522-4156
Bertrand Bearzatto  http://orcid.org/0000-0001-5193-6819

Patrice D. Cani http://orcid.org/0000-0003-2040-2448
Julien De Greef http://orcid.org/0000-0003-0200-1237
Vincent Haufroid http://orcid.org/0000-0001-5040-9806
Sébastien Jodogne http://orcid.org/0000-0001-6685-7398
Jean Cyr Yombi http://orcid.org/0000-0001-8424-2346
Leïla Belkhir http://orcid.org/0000-0002-1701-7584
Laure Elens http://orcid.org/0000-0002-0039-3583

## FUNDING

| Funder | Grant(s) | Author(s) |
| --- | --- | --- |
| Fondation Saint Luc | 2021-I4201010-221801 | Jean-Luc Balligand |
| | | Julien De Greef |
| | | Laurent Gatto |
| | | Jean Cyr Yombi |
| | | Leïla Belkhir |
| | | Laure Elens |
| Fonds De La Recherche Scientifique - FNRS | HC01020F | Patrice D. Cani |
| Fonds Spéciaux de Recherche | ARC 25/30-151 | Patrice D. Cani |
| Walloon excellence in life sciences and biotechnology | WELBIO-CR-2022A-02P | Patrice D. Cani |
| Fonds De La Recherche Scientifique - FNRS | EOS 40007505 | Patrice D. Cani |
| Fondation Saint Luc | FRC | Leïla Belkhir |
| Fonds Spéciaux de Recherche | 2021 | Laure Elens |

## AUTHOR CONTRIBUTIONS

Bradley Ward, Conceptualization, Data curation, Formal analysis, Investigation, Methodology, Resources, Validation, Visualization, Writing – original draft, Writing – review and editing | Laure B. Bindels, Formal analysis, Methodology, Software, Validation, Writing – review and editing | Jean-Luc Balligand, Conceptualization, Funding acquisition, Investigation, Methodology, Project administration, Supervision, Writing – review and editing | Bertrand Bearzatto, Data curation, Methodology, Writing – review and editing | Guido Bommer, Investigation, Supervision, Writing – review and editing | Patrice D. Cani, Conceptualization, Formal analysis, Investigation, Methodology, Project administration, Resources, Supervision, Validation, Writing – review and editing | Julien De Greef, Conceptualization, Data curation, Formal analysis, Funding acquisition, Investigation, Methodology, Project administration, Validation, Visualization, Writing – review and editing | Joseph P. Dewulf, Conceptualization, Investigation, Methodology, Writing – review and editing | Laurent Gatto, Conceptualization, Data curation, Formal analysis, Funding acquisition, Investigation, Methodology, Project administration, Software, Supervision, Validation, Writing – review and editing | Vincent Haufroid, Conceptualization, Funding acquisition, Investigation, Methodology, Project administration, Writing – review and editing | Sébastien Jodogne, Data curation, Investigation, Methodology, Writing – review and editing | Benoît Kabamba, Conceptualization, Data curation, Funding acquisition, Investigation, Methodology, Project administration, Resources, Supervision, Validation, Writing – review and editing | Sébastien Pyr dit Ruys, Conceptualization, Investigation, Methodology, Writing – review and editing | Didier Vertommen, Conceptualization, Investigation, Methodology, Writing – review and editing | Jean Cyr Yombi, Conceptualization, Funding acquisition, Investigation, Methodology, Project administration, Resources, Supervision, Writing – review and editing | Leïla Belkhir, Conceptualization, Funding acquisition, Investigation, Methodology, Project administration, Resources, Supervision, Validation, Writing – review and editing | Laure

Elens, Conceptualization, Data curation, Funding acquisition, Investigation, Methodology, Project administration, Resources, Supervision, Validation, Visualization, Writing – review and editing

## DATA AVAILABILITY

Host read depleted and trimmed raw data have been deposited in the European Nucleotide Archive (ENA) at EMBL-EBI under accession number PRJEB88674. Non-host depleted raw data are not available due to ethical and GDPR considerations.

## ETHICS APPROVAL

The study was conducted in accordance with the Declaration of Helsinki and approved by the Institutional Review Board (or Ethics Committee) of Cliniques universitaires Saint Luc (comité éthique hospital-facultaire) (protocol code 2021/30DEC/543; date of approval, 30 December 2021). Informed consent was obtained from all subjects involved in the study.

## ADDITIONAL FILES

The following material is available online.

### Supplemental Material

**Supplemental material (Spectrum02313-25-s0001.docx).** Supplemental methods, Table S1, and Fig. S1 and S2.

### Open Peer Review

**PEER REVIEW HISTORY (review-history.pdf).** An accounting of the reviewer comments and feedback.

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
