## [Reviewer comments · Microbiology Spectrum]

Microbiology Spectrum

Association of nasopharyngeal *Dolosigranulum pigrum* and *Corynebacterium* species with post-acute sequelae of SARS-CoV-2 (PASC) in a longitudinal cohort

Bradley Ward, Laure Bindels, Jean-Luc Balligand, Bertrand Bearzatto, Guido Bommer, Patrice Cani, Julien De Greef, Joseph Dewulf, Laurent Gatto, Vincent Haufroid, Sébastien Jodogne, Benoit Kabamba, Sébastien Pyr dit Ruys, Didier Vertommen, Jean Cyr Yombi, Leïla Belkhir, and Laure Elens

Corresponding Author(s): Laure Elens, Universite catholique de Louvain Louvain Drug Research Institute

Review Timeline:

Submission Date:	July 31, 2025
Editorial Decision:	October 15, 2025
Revision Received:	January 12, 2026
Accepted:	January 19, 2026

Editor: Jan Claesen

Reviewer(s): Disclosure of reviewer identity is with reference to reviewer comments included in decision letter(s). The following individuals involved in review of your submission have agreed to reveal their identity: Cindy M Liu (Reviewer #1)

Transaction Report:

DOI: <https://doi.org/10.1128/spectrum.02313-25>

Re: Spectrum02313-25 (**The Potential Protective Role of *Dolosigranulum pigrum* and *Corynebacterium* species for PASC Development**)

Dear Prof. Laure Elens:

Thank you for the privilege of reviewing your work. Below you will find my comments, instructions from the Spectrum editorial office, and the reviewer comments.

Thank you for submitting your research to Spectrum. Your paper has been evaluated by two independent Reviewers and they highlighted several comments and suggestions to help improve the manuscript. I would be happy to consider a revised version which addresses these comments in a point-by-point manner. Please pay particular attention to 1) providing sufficient details on the methods and study (design) used in the paper, and 2) make sure that the claims are supported by the data presented - which does not necessarily mean performing additional experiments but adjusting claims and terminology. For example a 'protective role of *D. pigrum* or *Corynes*' as suggested in the title is not based on experimental evidence, and similarly terms like 'beneficial' and 'protective' should be used with caution throughout, as pointed out by Reviewer 1.

Revision Guidelines

Sincerely,
Jan Claesen
Editor
Microbiology Spectrum

Reviewer #1 (Comments for the Author):

Reviewer comments

This is a post-infection nasopharyngeal microbiome study; therefore, the authors are studying the impact of COVID-19 (time point 1) and what post-infection microbiome is associated with PASC (time points 1 and 2). Based on the study design, interpretations, such as "Beneficial" and "protective", which are used in Abstract/Importance/Discussions are not supported by their findings; however, descriptions such as "commensal" may be more suitable.

Also, while healthy patients and patients with influenza were included in the study, with patients with COVID-19, it is unclear based on the current paper if and how those comparator groups were included in the analyses. It appears that there were limited analyses that included the comparator populations. The compositions for each group at each time point is not described in sufficient detail in the study. And it's unclear which timepoints are included in the analyses reported.

Lastly, there are also major concerns that the nasopharyngeal microbiome composition reported in this study differs substantially with what's known about adult nasopharyngeal microbiome.

Some specific comments are below:

Abstract

- Line 60: This statement needs to be revised since there is no data supporting lung damage prevention during acute stage.: "certain commensal species, including *D. pigrum* and *Corynebacterium* species, may protect against PASC by preventing excessive lung damage during acute disease."

- Line 70: Author states that "bacteria naturally found in the nose and throat (known as the nasopharyngeal microbiome)" - this is not accurate. Nasopharyngeal microbiome is microbiome of the nasopharynx, nasal microbiome is from the nose, and oropharyngeal microbiome is from the throat. The statement should be corrected.

- Line 75: "*Dolosigranulum pigrum* and *Corynebacterium* species, seemed to help protect against Long-COVID". This needs to be corrected - they are just associated with the group that does not develop long COVID

Results

- Line 121-123 (and supplementary table E1): The nasopharyngeal microbiome composition reported in this study varies substantially from published studies (De Boeck et al., 2017, etc) based on the dominant/abundant taxa reported. This substantial deviation is most likely due to how the metagenome data was processed and classified. This creates concern that the community data from this metagenome study differs significantly from current state of knowledge and will not be translatable/generalizable, and also impacts the other analyses in this paper, such as Diversity analyses.

- Line 164-173 (and the rest of the paper): The confounding between age and sex with prior vaccine status and PASC and COVID-19 severity were not really addressed in the analyses. Age and sex are known to be associated with microbiome composition.

- The results include a lot of details about the methods, but also somehow it's unclear which data sets are used in which analyses. Further clarifications are needed in the Methods and Results section are needed. I would recommend potentially clarifying the analyses in terms of baseline comparison across groups, baseline comparisons between disease severity and PASC, and also looking at the change with PASC. It is very difficult to understand the data analyzed and the details about the model(s) used.

Reviewer #2 (Comments for the Author):

The manuscript proves data for important topic - microbiota composition and long covid. I have only minor comments for the content.

Please always write the species name in italic.

Line 85 - *Streptococcus* are very often considered pathobionts, that's why I would not write that they protect from pathogens.

Line 112 - symptoms.

Line 118 Please don't use vague vocabulary like GENERALLY in scientific paper.

Line 128-133 Please refer to the proper figure - if you talk about the results they need to be reported in the manuscript

Line 166 The species where significantly upregulated in which condition?

The correlations at figure 5 unclear, need to be described better.

Reviewer comments

This is a post-infection nasopharyngeal microbiome study; therefore, the authors are studying the impact of COVID-19 (time point 1) and what post-infection microbiome is associated with PASC (time points 1 and 2). Based on the study design, interpretations, such as “Beneficial” and “protective”, which are used in Abstract/Importance/Discussions are not supported by their findings; however, descriptions such as “commensal” may be more suitable.

Also, while healthy patients and patients with influenza were included in the study, with patients with COVID-19, it is unclear based on the current paper if and how those comparator groups were included in the analyses. It appears that there were limited analyses that included the comparator populations. The compositions for each group at each time point is not described in sufficient detail in the study. And it's unclear which timepoints are included in the analyses reported.

Lastly, there are also major concerns that the nasopharyngeal microbiome composition reported in this study differs substantially with what's known about adult nasopharyngeal microbiome.

Some specific comments are below:

Abstract

- *Line 60: This statement needs to be revised since there is no data supporting lung damage prevention during acute stage.: “certain commensal species, including D. pigrum and Corynebacterium species, may protect against PASC by preventing excessive lung damage during acute disease.”*
-
- *Line 70: Author states that “bacteria naturally found in the nose and throat (known as the nasopharyngeal microbiome)” – this is not accurate. Nasopharyngeal microbiome is microbiome of the nasopharynx, nasal microbiome is from the nose, and oropharyngeal microbiome is from the throat. The statement should be corrected.*
-
- *Line 75: “Dolosigranulum pigrum and Corynebacterium species, seemed to help protect against Long-COVID”. This needs to be corrected – they are just associated with the group that does not develop long COVID*
-

Results

- *Line 121-123 (and supplementary table E1): The nasopharyngeal microbiome composition reported in this study varies substantially from published studies (De Boeck et al., 2017, etc) based on the dominant/abundant taxa reported. This substantial deviation is most likely due to how the metagenome data was processed and classified. This creates concern that the community data from this metagenome study differs significantly from current state of knowledge and will not be translatable/generalizable, and also impacts the other analyses in this paper, such as Diversity analyses.*

- Line 164-173 (and the rest of the paper): The confounding between age and sex with prior vaccine status and PASC and COVID-19 severity were not really addressed in the analyses. Age and sex are known to be associated with microbiome composition.
- The results include a lot of details about the methods, but also somehow it's unclear which data sets are used in which analyses. Further clarifications are needed in the Methods and Results section are needed. I would recommend potentially clarifying the analyses in terms of baseline comparison across groups, baseline comparisons between disease severity and PASC, and also looking at the change with PASC. It is very difficult to understand the data analyzed and the details about the model(s) used.
-

Global response

Causal/ “protective” wording and title

We thank the reviewers and editors for highlighting concerns regarding the use of causal terminology such as “protective,” “beneficial,” or “protective consortium.” We fully agree that, given the observational nature of our study, such terms may imply mechanistic effects that our data cannot directly support. In response, we have systematically revised the manuscript to ensure that all microbiome–PASC relationships are described strictly as **associations**.

Across the manuscript, wording such as “*protective*,” “*beneficial*,” “*protect against PASC*,” and “*protective consortium*” has been replaced with terms such as “*associated with lower PASC prevalence*,” “*commensal taxa associated with favourable outcomes*,” or “*candidate biomarkers*.” This includes revisions throughout the **title, running title, Abstract, Importance, Results, Discussion, and Figure 5 legend**.

Key changes include:

- **Title:**
Now: “Association of nasopharyngeal Dolosigranulum pigrum and Corynebacterium species with post-acute sequelae of SARS-CoV-2 (PASC) in a longitudinal cohort.”
- **Abstract / Importance:**
All statements that previously implied protective effects were rewritten to describe differences in abundance between individuals who recover and those who develop PASC, and to reference experimental data only as contextual evidence — not as conclusions from our study.
- **Discussion:**
Phrases implying protection or immune-mediated prevention have been removed or reframed. For example:
 - “suggesting a protective respiratory role” → removed
 - Mechanistic interpretations (e.g., lung-damage prevention) were rewritten to explicitly state that **our data remain associative**, while acknowledging supporting evidence from experimental studies as background context (See point-by-point response to reviewer 1 comment 1).
- **Figure 5 legend:**
The term “protective consortium” was replaced with “*commensal consortium associated with lower PASC frequency*.”

We believe that these changes ensure that the manuscript accurately reflects the **observational scope** of the study and avoids any implication of direct causality. Importantly, the scientific message and main conclusions remain unchanged: certain commensal taxa differ reproducibly between patients who develop PASC and those who recover, and these patterns are consistent with hypotheses generated in independent experimental literature.

Additionally, during the bioinformatic scrutiny employed in response to the comments from the reviewers, an error in assigning names was identified in the code used to generate figure 5, resulting in the deletion of a number of edges in the SparCC co-abundance network. This error has now been fixed and the resulting network now accurately reflects the data. Importantly, the additional edges now included in this accurate network do not affect our conclusions regarding *D. pigrum*, *C. accolens*, and other microbial species. The network has also been updated in terms of node size and colour (explained in the figure). Finally, exact node positioning has changed due to the stochastic nature by

which the network layout is generated, but this is a purely aesthetic change and does not affect the results. We thank the reviewers very much for causing us to delve back into the bioinformatic workflows and allowing us to spot this error before publication.

Point-by-point response to the reviewers

Reviewer 1

This is a post-infection nasopharyngeal microbiome study; therefore, the authors are studying the impact of COVID-19 (time point 1) and what post-infection microbiome is associated with PASC (time points 1 and 2). Based on the study design, interpretations, such as "Beneficial" and "protective", which are used in Abstract/Importance/Discussions are not supported by their findings; however, descriptions such as "commensal" may be more suitable.

We thank the reviewer for this important comment. We fully agree that, given the observational nature of our study and the post-infection sampling design, causal terminology such as "protective" or "beneficial" is not warranted. This concern was also raised by the editor, and in response we have undertaken a thorough revision of the manuscript to ensure that all interpretations remain strictly association-based.

Specifically, we have removed or rephrased every instance of causal or mechanistic language throughout the Abstract, Importance section, Results, and Discussion. Terms such as "*protective*," "*beneficial*," "*protective consortium*," or "*protect against PASC*" have been replaced with non-causal formulations, including "*associated with lower PASC prevalence*," "*commensal taxa associated with favourable outcomes*," "*candidate biomarkers*," or "*microbial signatures linked to PASC status*." These revisions ensure that the manuscript does not imply biological effects that our study design cannot support.

In addition, we have updated the title and running title to reflect an association-based framing:

- **Revised title:** "*Association of nasopharyngeal *Dolosigranulum pigrum* and *Corynebacterium* species with post-acute sequelae of SARS-CoV-2 (PASC) in a longitudinal cohort.*"
- **Revised running title:** "*Nasopharyngeal microbiome and PASC.*"

We believe these changes directly address the reviewer's concern and more accurately convey the scope and limitations of our findings, while maintaining the scientific value of the associations observed.

Also, while healthy patients and patients with influenza were included in the study, with patients with COVID-19, it is unclear based on the current paper if and how those comparator groups were included in the analyses. It appears that there were limited analyses that included the comparator populations. The compositions for each group at each time point is not described in sufficient detail in the study. And it's unclear which timepoints are included in the analyses reported.

We thank the reviewer for this comment. We agree that the role of the comparator groups may not have been sufficiently explicit in the narrative, and we are happy to clarify how they were used in the analyses (See point-by-point response to reviewer 1 comment 6). Healthy controls and influenza patients were fully included in all analyses aimed at comparing overall nasopharyngeal microbiome composition across groups. This includes the study cohort description, alpha-diversity and beta-diversity analyses at both timepoints, and the global co-abundance network, all of which used data from *all* participant groups at visits 1 and 2.

In contrast, analyses focusing on **COVID-specific questions** (COVID-19 severity, PASC development, and antibiotic exposure during COVID-19 hospitalization) necessarily included **COVID-19 patients only**, as these outcomes are not defined for influenza or healthy participants.

All participants were sampled at two timepoints, and the analyses used either both visits (for whole-cohort comparisons) or one visit at a time (for COVID-specific differential abundance analyses), as stated in the Results.

Lastly, there are also major concerns that the nasopharyngeal microbiome composition reported in this study differs substantially with what's known about adult nasopharyngeal microbiome.

Thank you for raising this important point. We agree that our nasopharyngeal microbiome profiles differ from what is typically described in healthy adult cohorts. However, this difference is biologically and clinically expected given the nature of our study population. A Detailed answer is given to reviewer's 4th specific comment (see below).

Some specific comments are below:

1. Line 60: This statement needs to be revised since there is no data supporting lung damage prevention during acute stage.: "certain commensal species, including *D. pigrum* and *Corynebacterium* species, may protect against PASC by preventing excessive lung damage during acute disease."

Thank you for pointing this out. We agree that our original wording could be interpreted as implying a causal or mechanistic effect on lung damage during the acute phase, which our observational data cannot support. We have therefore revised the sentence to remove any causal inference and to align it strictly with what our data show.

The revised sentence now reads: "Furthermore, the analysis of microbial co-associations identifies commensal species, including *D. pigrum* and *Corynebacterium* species, that are less abundant in patients who develop PASC, consistent with a potential protective role suggested by experimental studies but not proven by our observational data".

This wording avoids unsupported mechanistic claims while still acknowledging that experimental literature points to possible immunomodulatory roles of these taxa. Our manuscript now consistently uses an association-based framing throughout.

2. Line 70: Author states that "bacteria naturally found in the nose and throat (known as the nasopharyngeal microbiome)" - this is not accurate. Nasopharyngeal microbiome is microbiome of the nasopharynx, nasal microbiome is from the nose, and oropharyngeal microbiome is from the throat. The statement should be corrected.

Thank you for this helpful clarification. We agree that our original wording conflated distinct anatomical sites and did not accurately reflect the definition of the nasopharyngeal microbiome. In response, we have corrected the statement to be anatomically precise.

The revised sentence now reads: "This study highlights the importance of bacteria naturally found in the upper respiratory tract, particularly the nasopharynx (the nasopharyngeal microbiome), in shaping..."

This correction clearly distinguishes the nasopharynx from the nasal and oropharyngeal niches and ensures accurate terminology throughout the manuscript.

3. Line 75: "*Dolosigranulum pigrum* and *Corynebacterium* species, seemed to help protect against Long-COVID". This needs to be corrected - they are just associated with the group that does not develop long COVID Results.

Thank you for this comment. We agree that the original phrasing could be interpreted as implying a protective effect, which our observational data cannot support. We have therefore revised the sentence to describe the findings strictly as associations.

The corrected statement now reads: "...we found that certain commensal bacteria, *namely Dolosigranulum pigrum* and *Corynebacterium* species, were less abundant in individuals who developed Long-COVID, and more abundant in those who fully recovered."

This wording accurately reflects the association observed without suggesting causality.

4. Line 121-123 (and supplementary table E1): The nasopharyngeal microbiome composition reported in this study varies substantially from published studies (De Boeck et al., 2017, etc) based on the dominant/abundant taxa reported. This substantial deviation is most likely due to how the metagenome data was processed and classified. This creates concern that the community data from this metagenome study differs significantly from current state of knowledge and will not be translatable/generalizable, and also impacts the other analyses in this paper, such as Diversity analyses.

Thank you for raising this. We agree our nasopharyngeal profiles differ from healthy reference cohorts such as De Boeck et al. (2017). However, we believe this discrepancy is expected and biologically meaningful rather than a consequence of data processing or taxonomic classification artefacts.:

Clinical context explains deviation from healthy reference profiles

The comparison with De Boeck et al. is not directly like-for-like. Our cohort consists primarily of individuals with acute, often severe viral respiratory infections (severe influenza and severe COVID 19), some of whom had recent antibiotic exposure and hospital-level care. These conditions are well known to induce **strong upper airway dysbiosis** and expansion of opportunistic/pathobiont taxa, unlike healthy reference cohorts such as De Boeck et al. Studies of symptomatic respiratory viral infections consistently report enrichment of Staphylococcus, Klebsiella, and other opportunists, and severe influenza in particular has been shown to exhibit super-dominance of single pathobionts, consistent with our findings (Edouard *et al.*, 2018; Claassen-Weitz *et al.*, 2025; Hao *et al.*, 2025; Qin *et al.*, 2019; DeMuri *et al.*, 2017).

Methodological robustness ensures that the observed profiles are not artefacts.

We do not attribute the observed taxonomic patterns to methodological bias. Shotgun metagenomic data were processed using a conservative, strongly robust workflow, including stringent quality control and trimming, host read removal against GRCh38, taxonomic assignment using Kraken2 with Bracken abundance estimation, application of minimum read-depth thresholds, exclusion of low-yield samples, and contaminant detection using decontam (Timilsina *et al.*, 2025; Govender and Eyre 2022; Gao *et al.*, 2025). Alpha diversity and beta-diversity analyses were also performed using appropriate compositional methods with checks for dispersion effects. We therefore do not attribute the observed taxonomic shifts to methodological bias.

Use of community standards confirms accuracy and reliability.

To further ensure accuracy, balanced microbial community standards were included in every sequencing batch. As shown in Supplementary Figure E2, all expected species were correctly identified. While absolute abundance accuracy varied for a few taxa, relative abundance estimation was consistent. This is particularly relevant given that our primary analyses rely on relative abundance and log-fold-change comparisons rather than absolute quantification.

Misclassified reads were rare, largely confined to correct genus-level assignments and occurred at very low abundance, making them unlikely to influence downstream analyses or conclusions.

Taken together, the divergence from healthy nasopharyngeal microbiome profiles reflects the clinical severity and treatment context of the study population rather than limitations of the metagenomic pipeline. While these data are not intended to be generalized to healthy individuals, they are highly

relevant and translatable to understanding microbiome alterations in severe viral respiratory disease, which is the focus of this study.

5. Line 164-173 (and the rest of the paper): The confounding between age and sex with prior vaccine status and PASC and COVID-19 severity were not really addressed in the analyses. Age and sex are known to be associated with microbiome composition.

Thank you for this comment. We agree that age and sex are important potential confounders, both because they differ across clinical groups (severity, vaccination status, PASC) and because they are known to influence microbiome composition. To address this, we reran all relevant statistical models including age and sex as covariates.

Incorporating these covariates led to only minor changes in specific species-level results:

- Visit 1 PASC vs recovered (Figure 2):
C. tuberculostearicum is no longer significantly differentially abundant.
- Antibiotic-treated vs untreated at visit 1 (Figure 4):
C. tuberculostearicum, *S. pneumoniae*, and *C. acnes* are now significant, while *C. pseudodiphtheriticum* is not.

All other models remained unchanged in terms of significant species, although logFC and p-values were updated accordingly.

Importantly, **these adjustments do not alter the central conclusions of the study**: the taxa associated with PASC and the broader microbiome patterns remain consistent after controlling for age and sex.

6. The results include a lot of details about the methods, but also somehow it's unclear which data sets are used in which analyses. Further clarifications are needed in the Methods and Results section are needed. I would recommend potentially clarifying the analyses in terms of baseline comparison across groups, baseline comparisons between disease severity and PASC, and also looking at the change with PASC. It is very difficult to understand the data analyzed and the details about the model(s) used.

Thank you for this comment. We agree that clarity regarding which subsets of participants were used for each analysis is essential. In response, we have revised both the Methods and Results sections to explicitly state, for every analysis, (i) the sub-cohort used, (ii) the statistical model applied, and (iii) the confounders included.

After discussion within the research team, we retained the original analytical structure—

- **Visit 1 analyses:** COVID-19 severity and PASC development
- **Visit 2 analyses:** PASC status
- **Longitudinal analyses:** changes related to PASC
- **Antibiotic exposure analyses:** among hospitalized patients

—but we have substantially clarified these distinctions throughout the text.

The revised sections now make it clear which dataset corresponds to each analysis and how each model was specified, ensuring the analytical workflow is straightforward to follow.

Reviewer 2

1. Please always write the species name in italic.

Thank you for this remark. All bacterial species names have now been corrected to italic format throughout the manuscript.

2. Line 85 - Streptococcus are very often considered pathobionts, that's why I would not write that they protect from pathogens.

Thank you for this important observation. We fully agree that *Streptococcus* species are frequently considered pathobionts in the upper respiratory tract and therefore should not be described as “protecting against pathogens.” Our intention was instead to describe the dominant genera typically found in a healthy adult nasopharyngeal microbiome, and then to refer more generally to the concept of community balance and colonization resistance.

The sentence has therefore been rewritten for accuracy and clarity as follows: “In healthy adults, the nasopharyngeal microbiome is typically dominated by genera such as *Corynebacterium*, *Dolosigranulum*, *Moraxella*, *Streptococcus* and *Haemophilus*; a balanced community is thought to contribute to mucosal immune maturation and colonization resistance against respiratory pathogens”.

This revision avoids suggesting a protective role for *Streptococcus* species and reflects current understanding of nasopharyngeal ecology.

3. Line 112 - symptoms.

Thank you for pointing this out. The typo at this line has now been corrected.

4. Line 118 Please don't use vague vocabulary like GENERALLY in scientific paper.

Thank you for highlighting this point. We agree that vague terms such as “generally” should be avoided in scientific writing. We have therefore replaced the sentence with precise wording: “Severe COVID-19 and PASC patients had a higher mean age, a higher proportion of male patients, a higher mean BMI, a higher prevalence of smoking and drinking, and a lower prevalence of full vaccination than those with mild cases or full recovery”.

We believe this revision provides specific comparative information without using imprecise vocabulary.

5. Line 128-133 Please refer to the proper figure - if you talk about the results they need to be reported in the manuscript

Thank you for this comment. In this section, no figures were generated because the alpha-diversity analyses were based solely on per-sample diversity measures followed by Kruskal–Wallis and Wilcoxon tests to evaluate differences across patient groups. These analyses do not produce a figure by default.

To improve clarity, we have now added the corresponding statistical results directly into the text at this point in the manuscript so that readers can see the numerical outcomes even in the absence of a figure.

6. Line 166 The species where significantly upregulated in which condition?

Thank you for this comment. We agree that the original phrasing was unclear regarding the direction of differential abundance. We have therefore rewritten the sentence to explicitly state the condition in which each species was more or less abundant: “At visit 2, 4 species were found to be significantly upregulated: *C. propinquum* ($q = 8.41e-06$, $\text{LogFC} = 2.93$), *Cutibacterium granulosum* ($q = 4.22e-05$, $\text{LogFC} = 1.57$), *C. pseudodiphtheriticum* ($q = 1.34e-03$, $\text{LogFC} = -1.16$), and *C. accolens* ($q = 7.87e-03$, $\text{LogFC} = -1.54$).” **replaced with** “At visit 2, four species were found to differ significantly in their abundance: *C. propinquum* ($q = 2.07e-04$, $\text{LogFC} = 2.43$) and *Cutibacterium granulosum* ($q = 2.07e-04$,

LogFC = 1.22) were more abundant in PASC, whilst *C. pseudodiphtheriticum* ($q = 1.42e-05$, LogFC = -1.73), and *C. accolens* ($q = 5.54e-03$, LogFC = -1.68) were less abundant.”

This revised wording clearly specifies the condition associated with increased or decreased abundance.

7. The correlations at figure 5 unclear, need to be described better.

Thank you for raising this point. We agree that the correlations shown in Figure 5 required clearer explanation. To improve interpretability, we have expanded the **Methods section** to provide a detailed description of how the co-abundance network was constructed (species filtering, SparCC correlation estimation, bootstrap significance testing, FDR correction, and thresholds for inclusion).

Specifically : “To infer global co-abundance patterns between species, we first restricted the dataset to species present in at least 10% of all samples (all patient groups, visit 1 and visit 2). We then computed SparCC correlations using species raw counts (SpiecEasi::sparcc). Statistical support for each correlation was assessed using 1,000 bootstrap resamples, and p-values were adjusted for multiple testing using the Benjamini–Hochberg procedure. We retained taxon correlation pairs with false discovery rate (FDR) < 0.01 and an absolute correlation coefficient $|r| \geq 0.3$ to construct an undirected co-abundance network. Nodes represent species and edges represent significant correlations; positive correlations are depicted in green and negative correlations in red.”

In the Results section, Line 205-208 “In order to assess the complex microbial co-associations found within the microbiome, pairwise SparCC compositionally robust correlation estimates were calculated, followed by bootstrap significance estimates. A similarity network was then created based on significant (FDR < 0.01), robust (absolute association > 0.3) pairwise associations.” **was simplified to the corresponding text**: “We next examined significant co-abundance relationships between prevalent nasopharyngeal species using a SparCC-based association network (Figure 5)”. This avoids overloading the Results with methodological detail while clarifying the purpose of the analysis.

We have also **rewritten the Figure 5 legend** to explicitly explain what the network represents (nodes, edges, positive vs negative associations), and we have expanded the **Supplementary Methods** with a full description of the network generation process.

Together, these changes make the correlations and their interpretation significantly clearer for the reader.

Re: Spectrum02313-25R1 (**Association of nasopharyngeal *Dolosigranulum pigrum* and *Corynebacterium* species with post-acute sequelae of SARS-CoV-2 (PASC) in a longitudinal cohort**)

Dear Prof. Laure Elens:

Thanks for carefully addressing the Reviewers' comments, improving the manuscript. I would hereby like to congratulate you on the acceptance of your paper for publication in Spectrum!

Your manuscript has been accepted, and I am forwarding it to the ASM production staff for publication. Your paper will first be checked to make sure all elements meet the technical requirements. ASM staff will contact you if anything needs to be revised before copyediting and production can begin. Otherwise, you will be notified when your proofs are ready to be viewed.

Sincerely,
Jan Claesen
Editor
Microbiology Spectrum